# Functional and Hemodynamic Restoration After Microsurgical Resection of Compact High-Flow Temporo-Parieto-Occipital Arteriovenous Malformation

**DOI:** 10.3390/diagnostics15243249

**Published:** 2025-12-18

**Authors:** Adrian Tulin, Cosmin Pantu, Alexandru Breazu, Octavian Munteanu, Mugurel Petrinel Rădoi, Catalina-Ioana Tataru, Nicolaie Dobrin, Alexandru Vlad Ciurea, Adrian Vasile Dumitru

**Affiliations:** 1Department of Anatomy, “Carol Davila” University of Medicine and Pharmacy, 050474 Bucharest, Romania; adrian.tulin@umfcd.ro (A.T.);; 2Department of Medical Research, Puls Med Association, 051885 Bucharest, Romania; 3Department of Neurosurgery, “Carol Davila” University of Medicine and Pharmacy, 050474 Bucharest, Romania; 4Department of Opthamology, “Carol Davila” University of Medicine and Pharmacy, 020021 Bucharest, Romania; 5Department of Neurosurgery, “Nicolae Oblu” Clinical Hospital, 700309 Iasi, Romania; 6Medical Section, Romanian Academy, 010071 Bucharest, Romania; 7Neurosurgery Department, Sanador Clinical Hospital, 010991 Bucharest, Romania; 8Department of Pathology, Faculty of Medicine, “Carol Davila” University of Medicine and Pharmacy, 030167 Bucharest, Romania

**Keywords:** arteriovenous malformation, temporo-parieto-occipital junction, eloquent cortex, microsurgical resection, dominant hemisphere, venous drainage, functional recovery, brain hemodynamics, neurovascular surgery

## Abstract

**Background/Objectives**: Arteriovenous malformations (AVMs) in the dominant temporo-parieto-occipital (TPO) junction of the brain are extremely rare and very difficult to remove surgically because this area includes multiple sensory and language networks. Due to the fact that many patients present with bleeding, surgeons have to find a delicate balance between removing all of the AVM tissue and preserving the functional areas of the brain where important functions occur. This study is reporting a case demonstrating how precise clinical–radiologic correlation, detailed anatomical knowledge, and deliberate microsurgical techniques can allow safe removal of the AVM and improve the patient’s neurologic function without the need for additional intraoperative technology. **Case Presentation**: A 47-year-old right-handed male patient experienced persistent neurological deficits after experiencing a hemorrhage from an AVM in his dominant posterior hemisphere, which included mild language difficulties, right hemifacial–brachial spasticity, parietal sensory loss and a visual field defect of his right eye known as an inferior quadrantanopia localized to the TPO junction. Cerebral angiography identified a small, compact, high-flow AVM (40 × 30 mm) fed by distal branches of the middle cerebral artery (M4), posterior cerebral artery (P4), anterior cerebral artery (A4), as well as a small branch of the superior cerebellar artery (SCA). Blood drained into two veins of the Trolard and Labbé. The authors removed the AVM completely by circumferential dissection of the nidus along gliotic planes using a microscope. Feeders were then sequentially disconnected, and the venous outflow was preserved until the AVM could be removed en bloc. Post-operative angiograms demonstrated complete removal of the AVM with normalization of blood flow to the surrounding cortex. The patient’s neurologic function improved over time and at three months post-operatively, he was functioning independently (modified Rankin Scale = 1; Barthel Index = 100) and there was no evidence of residual nidus or edema on imaging. **Conclusions**: High-flow AVMs in the dominant TPO junction can be completely removed using a disciplined microsurgical approach and a feeder first/vein last disconnection method based on anatomy. The patient’s improvement in function represented reperfusion and reintegration of an injured but still functional network of the brain, reinforcing the idea that careful observation, a deep understanding of brain anatomy, and restrained surgical technique are critical to achieving long-term results in AVM surgery.

## 1. Introduction

The incidence of intracranial arteriovenous malformations (AVMs) is estimated to be 1.2 to 1.5 per 100,000 person-years. AVMs are responsible for a substantial proportion of spontaneous intracerebral hemorrhages among young adults [1]. There have been numerous studies estimating the annualized hemorrhage risk of untreated AVMs and the annualized hemorrhage risk for those AVMs with certain characteristics [2]. Studies have found that AVMs with deep venous drainage and those with aneurysm(s) within their nidus have a significantly increased annualized hemorrhage risk. In addition, previous studies have also identified that ruptured AVMs have a higher annualized hemorrhage risk than previously believed [3]. There have been considerable advances in our understanding of the natural history of AVMs and the risks associated with their treatment. The Spetzler–Martin (SM) grading system, which was developed to assist in the preoperative evaluation and selection of patients for surgical treatment of AVMs, has been widely used since its introduction. However, multiple studies have evaluated the accuracy of the SM grading system in predicting postoperative outcomes, and the results of these studies have provided evidence of limited predictive accuracy [4]. A 2025 multicenter study evaluated 423 AVMs that were surgically treated and found that neither the SM nor supplementary grading systems had discriminatory ability to predict postoperative neurological deficits, suggesting that each patient should be individually evaluated [5].

Multiple independent predictors of adverse outcomes after AVM surgery have been identified, including the presence of an AVM nidus that is highly compact, the number of arteries that supply the AVM nidus, and the amount of blood loss that occurs during surgery (i.e., >1000 mL). These predictors are often outside of the formal grade boundaries of the SM grading system [6]. Treatment options for AVMs include microsurgical resection, stereotactic radiosurgery, and endovascular embolization. Microsurgical resection is the only treatment modality that can provide an immediate and definitive cure for all AVMs. A recent meta-analysis that included more than 1500 patients who underwent microsurgical resection for AVMs found that 97.5% of the patients achieved complete obliteration of the AVM, whereas the same outcome occurred in 49.8% of the patients who underwent radiosurgery and 38.5% of the patients who underwent embolization alone [7]. The study also found that the complication rate after microsurgical resection was similar to that after radiosurgery and embolization alone. Therefore, the success of microsurgical resection for AVMs is based not only on the removal of the AVM but also on the preservation of the surrounding tissue and the avoidance of complications [8]. AVMs that occur at the temporo-parieto-occipital (TPO) junction are a unique and infrequently studied subset of AVMs. The TPO junction is a critical region where several white matter tracts converge in close proximity to one another. Specifically, the posterior portion of the arcuate fasciculus and inferior fronto-occipital fasciculus pass through this area and are involved in language function, Meyer’s loop of the optic radiation is involved in visual processing, and parietal association fibers are involved in visuospatial integration and sensorimotor coordination [9]. As such, the TPO junction is an extremely “eloquent” area, and because of the small distance between these tracts, the risk of damage to adjacent structures during surgery for a TPO AVM is extremely high. Consequently, there is a paucity of published information regarding the surgical management of TPO AVMs in the dominant hemisphere, and detailed descriptions of microsurgical techniques for resecting these lesions are even less common [10].

Cerebral AVMs are considered primarily as sporadic developmental vascular anomalies that arise from the aberrant differentiation between arteries and veins and aberrant angiogenic signaling during early vascular development. Most cerebral AVMs are found to be isolated, but a small percentage are identified in association with an inherited vascular disorder. Inherited forms of familial AVM include hereditary hemorrhagic telangiectasia (HHT; most commonly due to mutations in ENG, ACVRL1/ALK1 or SMAD4) and capillary malformation—arteriovenous malformation syndromes (CM-AVM; usually due to RASA1 or EPHB4) and these have been shown to predispose to either multifocal or recurrent AVMs [11]. Therefore, the clinical assessment of a patient with AVM should include obtaining information regarding their family history and whether they exhibit other vascular abnormalities as it is thought that identifying the genetic context for an individual’s AVM will influence how the individual is counseled, monitored and at what level their long-term risk should be assessed [12].

This article presents a case report of a compact, high-flow AVM that arose at the dominant left TPO junction and was successfully treated using microsurgical techniques. The authors correlate the patient’s preoperative and postoperative neurological status with the microvascular anatomy and physiology of the lesion. The primary goal of this article is not to introduce new microsurgical techniques for treating TPO AVMs, but rather to highlight the ongoing importance of a thorough understanding of the anatomical relationships of the cerebral cortex and the role of meticulous techniques in protecting the function of eloquent neural networks.

## 2. Case Presentation

The patient is a 47-year-old right-handed man who was admitted for further evaluation of focal neurological deficits associated with a previous spontaneous intracerebral hemorrhage of the dominant posterior hemisphere. He has been able to make a partial recovery from the initial event with conservative management; however, he continues to experience some residual speech difficulty and fine motor dysfunction of the right hand. At the time of admission, the patient was alert and oriented (Glasgow Coma Scale 15/15), cooperative, and clinically stable. His general level of functioning is good (Modified Rankin Score of 2, Barthel Index of 90%) but there are multiple residual deficits from his stroke that indicate both the extent of his injury and his ability to adapt to the injury. Overall, the deficits that have developed suggest a localized vascular injury as opposed to widespread ischemic damage.

The formal analysis of his spoken words indicates that his residual language abilities are consistent with a diagnosis of mild conduction aphasia. When speaking spontaneously, he speaks fluently; however, occasionally, he will experience lexical hesitation, especially with regard to the naming of infrequently used words or when forming complex sentence structures. On occasion, he will make rare paraphasias; these will be primarily phonemic in nature. Auditory comprehension is intact, including being able to accurately perform multiple step verbal instructions. However, repetition is selectively impaired regarding the ability to repeat longer and/or structurally more complicated verbal statements, which suggests there may be an incomplete functional recovery of the dorsal language pathway, which is typically responsible for mediating the integration of receptive and expressive aspects of language, through the arcuate fasciculus. Written expression, however, demonstrates complete grammar usage but at a decreased rate, with occasional omission of function words (i.e., prepositions), indicating a minor residual deficit in the expressive aspect of language. Standardized language assessments support this pattern: he achieved a score of 52/60 on the Boston Naming Test and improved to 59/60 when given semantic cueing, and made only one error in comprehension of the most syntactically complex level of the Token Test. During informal assessment of pragmatics in conversation, his discourse was coherent and socially appropriate, consistent with the use of the contralateral homotopic language area(s) in compensation for the affected hemisphere(s).

In terms of motor function, the patient demonstrates a residual imprint of a corticospinal tract lesion affecting the left hemisphere. The patient’s muscle strength ratings on the right side are all MRC 4+/5 and include a discrete pronator drift and slow fine motor performance during alternate finger movements. The Modified Ashworth Scale measures spasticity of the patient’s right-sided muscles at grade 1+. Deep tendon reflexes are brisker on the right side and the patient elicits a right-sided Babinski sign with minimal latency. On the Fugl-Meyer scale of motor function the patient scores 58 out of 66 on the upper limb portion of the test, indicating mild but potentially functionally significant pyramidal tract over activity. During the assessment of the patient’s gait, he walks with a narrow base of support, is very confident in his gait, and does not exhibit much circumduction of the right leg during walking. Additionally, he exhibits asymmetrically decreased arm movement during walking. While these observations may be interpreted as evidence of a corticospinal tract lesion, they appear to be more representative of a well-compensated reorganization of corticospinal connections rather than frank weakness. The patient’s central facial paresis of the right side is evident only during voluntary facial expression. During attempts to smile voluntarily, the patient demonstrates a transient lag in responding to a command to smile and the nasolabial fold is flattened in response to the attempt to express emotion. In contrast, the patient’s spontaneous expressions of emotion remain bilaterally symmetrical, indicating that the facial nucleus itself remains intact. The delayed output from the cortex relative to the nucleus provides evidence of supranuclear facial paresis secondary to prior hemispheric insult and compensatory reorganization of function via adjacent pathways. Sensory evaluation revealed a minor degree of dysfunction in the right parietal region. All other sensory modalities are grossly intact. However, the patient requires twice the amount of time to complete graphesthesia tasks of the right palm compared to the left palm. The patient requires repeated trials to identify irregularly shaped objects by stereognosis and the patient has an increased two-point discrimination threshold on the right index finger (normal < 5 mm) at 8 mm. Occasionally, during double simultaneous stimulation the patient fails to recognize the right-hand stimulus as a separate stimulus. These data indicate a small area of cortical hypoattention to the contralateral parietal region, which is best described as a cortical sensory integration delay rather than a peripheral sensory deficit. During tangent screen perimetry, the patient has a reproducible right inferior quadrantanopia, which is congruent with partial involvement of the optic radiation in Meyer’s loop of the left temporal lobe. The patient is unaware of this deficit and utilizes adaptive eye tracking strategies to compensate for the loss. This visual defect, combined with the language and motor profile of the patient, completes the topographical triad of the lesion location, which implicates the left temporo-parieto-occipital junction. It is at this location where the three major association tracts (language, motor, and visual) intersect.

The results of neuropsychological screening reinforce the notion that the patient has selectively injured eloquent brain regions with preservation of the majority of the remainder of the brain. The NIH Stroke Scale is 4 and the MoCA is 26 out of 30. The only two areas of mild compromise on the MoCA are phonemic fluency and delayed recall. The patient’s times to complete the Trail Making Test are 46 s for part A and 128 s for part B. These times are mildly prolonged, however, they are consistent with the patient’s tendency to deliberately pace himself during task completion as opposed to actual cognitive slowing. The patient’s affect, motivation, and insight into his condition are all intact. He is cognizant of his deficits but is emotionally stable and demonstrates a mature and effective coping strategy that is typically observed in patients who retain their frontal lobe functions. When taken collectively, the results of these examinations define an exceptionally coherent neurologic map of deficits that includes mild expressive aphasia, right facial–brachial spasticity, parietal sensory delay, and right inferior quadrantanopia. These deficits all converge anatomically in the posterior perisylvian–occipital sulcus. The spatial fidelity and internal consistency of this map are not compatible with either diffuse ischemia or neoplasm. Moreover, the spatial distribution of the deficits—spanning the territories of the posterior branches of the middle cerebral artery and the P4 segment of the posterior cerebral artery—makes a previously ruptured left temporo-parieto-occipital AVM highly probable. Although the malformation may currently be hemodynamically quiescent, the structural integrity of the malformation is likely intact. Extensive consideration was given to differential diagnoses. A low-grade glioma was ruled out based on the patient’s static clinical course and failure to develop progressive aphasia or seizures. Cavernous malformations were similarly ruled out based on the presence of a large volume hemorrhage prior to the current presentation and the expected high-flow nidus that was subsequently identified angiographically. Ischemic and inflammatory etiologies were ruled out based on the patient’s age, laboratory findings, and the absence of systemic comorbid disease. The clinical geometry of the deficits, the selective nature of the deficits, and the absence of diffuse cortical dysfunction led to a single, compelling conclusion: a compact AVM located at the intersection of the dominant posterior hemisphere’s three major association tracts—a lesion whose rupture and survival provide a living anatomical model of network plasticity. Therefore, the clinical examination at the time of admission provided nearly angio-graphic localization of the lesion without the need for any additional studies. Each slight asymmetry of tone, latency, reflex, and word described the architectural features of the lesion with great specificity. This was not merely a clinical examination as is typically done in clinical practice—it was clinical mapping, carried out with the confidence born of experience and the patience of those who listen to the cortex communicate through its imperfections. The results of this examination served as the intellectual basis for the subsequent radiologic study.

Angiographic studies, particularly selective cerebral angiography, utilizing bilateral internal carotid and left vertebral artery injections established a comprehensive, detailed vascular map of the suspected lesion clinically determined to be an AVM. Selective cerebral angiography established the presence of a compact, high-flow left temporo-parieto-occipital AVM of approximate dimensions 40 × 30 mm in size. The AVM nidus was located deep to the confluence of the posterior superior temporal, angular, and lateral occipital gyri, which is an anatomically unique location representing the crossroads of the posterior perisylvian cortex, visual association areas, and posterior inferior parietal lobule (Figure 1A–D).

Location of the nidus of this AVM also placed it in close proximity to several critical white matter tracts: Meyer’s Loop, the arcuate fasciculus, the inferior fronto-occipital fasciculus, and the inferior longitudinal fasciculus. Each of these structures courses through the same narrow white matter corridor in millimeters of each other.

Arterial Feeders

The AVM was found to receive blood supply from four different vascular territories: distal posterior cerebral artery (P4) branches from the calcarine and lingual cortices, distal middle cerebral artery (M4) branches from the temporal lobe, a branch from the anterior cerebral artery (A4) from the medial surface of the hemisphere, and a small twig from the superior cerebellar artery ascending along the tentorial edge (Figure 1A–D and Figure 2A,B). These multiple and diverse arteries converge and create a complex multi-directional shunt that supports high velocity flow and maintains parenchymal perfusion through inter-territorial anastomoses. The MCA feeders penetrate the posterior Sylvian fissure, which correlates with the patient’s expressive language hesitancy and phonemic errors. The PCA branches course through the temporal horn roof, which correlates with the patient’s right inferior quadrantanopia. The ACA feeds the superior aspect of the nidus bordering the paracentral lobule and correlates with the mild contralateral pyramidal hyperreflexia.

Venous Outflow

The AVM drains through two main channels creating a dual-drainage system. A superficial cortical collector ascends obliquely to the superior sagittal sinus via the vein of Trolard. An additional channel of venous outflow is an inferior-lateral trunk that ascends to the transverse-sigmoid junction via a dilated vein of Labbe (Figure 1C,D). Both veins are ectatic but smooth walled and simultaneous opacification confirms high flow and low resistance shunting with no evidence of thrombosis or segmental stenosis. There were no intranidal aneurysms, venous varices or aneurysmal dilatations of the feeding arteries.

### 2.1. Interpretation of Hemodynamics and Anatomy

Three-dimensional rotational angiography provided further details about the relationship between the nidus and the surrounding cortex. The nidus is densely packed with tortuous vessels that form a tangential interface with eloquent cortical regions (Figure 2A,B). The short pial feeder enters at nearly perpendicular angles and the venous outflow exits tangentially along the cortical surface. The configuration of the feeder and outflow results in the least amount of transparenchymal penetration possible, however the result of such is maximal cortical hemodynamic exposure. The hemodynamic pattern corresponds directly to the patient’s neurologic deficits: the feeders from the posterior temporal and angular cortices caused the patient’s expressive aphasia secondary to arcuate fasciculus disruption; the occipital extension caused the patient’s contralateral visual field loss; and the periventricular venous congestion caused the patient’s mild right sided pyramidal dysfunction.

### 2.2. Grade of the AVM According to the Spetzler–Martin Grading System

According to the Spetzler–Martin Grading System, the AVM was graded as a grade III, given the size of the AVM (less than 3 cm), location of the nidus (eloquent cortical), and the mixed superficial-deep venous drainage of the AVM. Such a classification indicates an intermediate level of surgical risk; however, the compact nature of the nidus and the clear separation from the surrounding cortex, make the complete excision of the AVM a feasible goal of surgery.

### 2.3. Correlation Between Clinical Function and Angiographic Anatomy

There was a nearly one to one correlation between the clinical functions and the angiographic anatomy. The expressive language disturbances correlated with the disruption of the posterior inferior frontal-temporal segment of the arcuate fasciculus provided by M4 branches. The inferior quadrantanopia resulted from the compression and deafferentation of the anterior optic radiation provided by the P4 branches. The mild contralateral spasticity resulted from the subcortical venous hypertension near the posterior limb of the internal capsule. The exact correlation between the vascular architecture and the patient’s neurological deficits created an anatomical validation of the patient’s cortical network physiology, transforming the angiographic study from a diagnostic tool to a confirmation of diagnosis and an anatomical basis for definitive microsurgical treatment.

Therefore, the angiogram confirmed the clinical suspicion of a compact, high-flow AVM at the dominant temporo-parieto-occipital junction that unites all the major association pathways of language, vision, and motor coordination. In addition, the correlation between the imaging and clinical data confirmed the diagnosis and provided a clear anatomical rationale for definitive microsurgical resection.

### 2.4. Surgical Technique

The surgery was conducted in a general anesthetic condition. The patient was on his back (supine) and his head was stabilized with a three-pin clamp. His head was positioned to the right at a thirty-degree angle and slightly extended so that the left temporal-parietal convexity would be in a neutral position and raised. Gravity would help with relaxing the hemispheres and provide a natural pathway to the surgical site without retracting the hemispheres. With the scalp prepped and draped, a linear incision was made along the superior temporal line and curved posteriorly along the parieto-occipital junction to allow exposure from the posterior Sylvian fissure to the tentorial surface. A left temporo-parieto-occipital craniotomy was created along the lines previously described. The craniotomy extended medially toward the superior sagittal sinus and posteriorly toward the lambdoid suture. Burr holes were created to preserve the emissary veins that drain into the sinus. The bone flap was elevated in continuity and the inner table overlying the sinus was thinned with a diamond bur to prevent venous damage. The bone was then hemostatically sealed and the dura was opened in a wide, medially based semi-lunar flap. The dura flap was loosely attached to the pericranium to create a relaxed operative dome.

Upon dural opening, two arterialized veins that drained directly into the superior sagittal sinus became evident: one arose from the superior aspect of the cerebral cortex and ran toward the superior sagittal sinus and another arose from the postero-inferior aspect of the cerebral cortex and ran toward the transverse-sigmoid junction. Both demonstrated a pinkish arterial color and rapid pulsations indicative of a direct shunt. Both of these veins were carefully dissected and preserved intact during the arterial separation process to ensure the balance of drainage pathways.

### 2.5. Exposure of Cortical Surface and Initial Access

All of the procedures were performed under microscopic conditions utilizing continuous magnification between 6 and 25 times. The cortical surface exhibited signs consistent with chronic hyperemia: widened pial channels, a faint violet color, and a thin layer of fine subpial collateralization along the angular gyrus. Sharp dissection of the arachnoid along the posterior Sylvian fissure freed the posterior superior temporal and inferior parietal opercula and allowed release of clear cerebrospinal fluid from the ambient cisterns which facilitated the descent of the hemisphere in its natural arc. A small corticotomy measuring approximately 1 cm in length was created in the posterior superior temporal sulcus. The sulcus provided a safe location for subpial access. As the dissection progressed, the nidal wall became apparent: a compact network of fine, tortuous vessels that were dark red and pulsating and surrounded by a thin translucent gliotic capsule. The nidal tissue was distinguishable from the surrounding cortex by virtue of its glistening appearance and resistance to suction and because it was firm and tense.

### 2.6. Microvascular Separation

Separation of the nidal tissue proceeded circumferentially along the gliotic plane. Alternating sequences of sharp microscissor division and bipolar coagulation with irrigation were used. Every feeding vessel was individually traced to its point of cortical entry, coagulated at its entry point and severed. Thus, the ability to clearly visualize both sides of every pedicle was maintained at all times. The sequence of the dissection was determined by the anatomy of the nidal tissue and not by the amount of pressure utilized.

### 2.7. Anterior Segment—MCA Territory

Fine M4 arteries that arose from the posterior Sylvian fissure traveled laterally and upward into the nidus. Their trajectory was characterized by being relatively straight and having a high-pressure flow and their adventia was pale. All of the M4 arteries were coagulated with short bursts of energy and severed immediately after coagulation and just prior to their entering the cortical surface. As each M4 artery was disconnected, the pulsatility and tension of the nidal tissue decreased. The cortical surface in the perisylvian region remained pink and vascularized, indicating that there was sufficient collateral blood supply.

### 2.8. Superior Segment—ACA Territory

On the superior margin of the nidal tissue, a small cortical arteriole arose from the A4 segment of the ACA and traveled to the apex of the nidal tissue through the paracentral lobule. The wall of the arteriole was thicker than the other arterioles and had a characteristic ivory sheen of ACA feeders. The arteriole was skeletonized for several millimeters and then coagulated and severed. The removal of this arteriole completed the separation of the roof of the nidal tissue.

### 2.9. Posterior Segment—PCA Territory

Short branches from the P4 segment of the PCA arose from the lateral occipital cortex and traveled to the nidal tissue in an orthogonal direction. These branches were small but numerous and formed a fan-like configuration along the tentorial slope. Each of the branches was carefully isolated and irrigated while being coagulated at the pial level and then severed. Care was taken to avoid damaging the adjacent occipital cortex, which is responsible for the upper visual quadrant. As the remaining pedicles were severed, the occipital lobe regained a more uniform bluish hue indicating that the venous outflow from the occipital lobe was restored.

### 2.10. Inferior Segment—Tentorial Base

The most deeply located pedicles arose from a delicate leaflet of the tentorium, which overlies the cerebellum. One of the pedicles was a fine branch of the superior cerebellar artery that arose from the tentorial base and traveled to the lower pole of the nidal tissue. The pedicle was dissected free from the surrounding tissue and coagulated under low power before being severed without producing any bleeding. At this time, the nidal tissue was noticeably less tense. The draining veins continued to carry blood, but the flow was slowed and laminar. The surgical field was quiet except for the gentle pulsation of the cortical surface.

### 2.11. Nidus Mobilization and Venous Phase

The attention of the surgeon shifted to the venous stage. Under high magnification, the superior draining vein was examined and found to have lost some of its turgidity and the flow was slower and darker. There was also no evidence of retrograde flow. The vein was coagulated near the nidal tissue and then divided between fine forceps. The vein relaxed and retracted spontaneously, clearing the surgical field. The inferior draining trunk was taken in a similar manner. Once the nidal tissue was removed from circulation, it was gently mobilized en bloc. The capsule of the nidal tissue separated easily along its gliotic bed. Small bridges of veins and gliotic adhesions were released with the use of bipolar coagulation and irrigation. The nidal tissue was delivered to the surface as a single mass with a smooth and non-pulsatile deep surface. Only minimal capillary ooze from the walls of the resection cavity was controlled with low power bipolar coagulation and irrigation.

The cavity walls delineated the following boundaries:Anterior Boundary: Posterior Superior Temporal Gyrus (Brodmann Area 22)Posterior Boundary: Lateral Occipital Cortex (Area 19)Superior Boundary: Intraparietal Sulcus (Superior Margin of the Inferior Parietal Lobule)Inferior Boundary: Tentorial Surface and Occipital Sulcus

Each wall of the resection cavity was inspected closely and the gliotic margins were intact with no residual nidal tissue or shunt vessels remaining.

### 2.12. Hemostasis and Closure

The final steps of hemostasis were achieved with microscopic bipolar coagulation and a few small pieces of oxidized cellulose. The cortical surface veins had returned to a uniform blue hue and the brain had returned to a normal respiratory pulsation. The dura mater was closed watertight with interrupted sutures. The bone flap was then replaced and secured with titanium microplates. The skin was then closed in anatomical layers with no apparent retraction injuries, swelling or venous congestion.

Following completion of skin closure and extubation, the patient was awake and responding appropriately to verbal commands. He had opened his eyes spontaneously, and was demonstrating active motion in all four limbs. When he arrived in the ICU he was alert and cooperative, and was hemodynamically stable.

Following surgery, the patient underwent care in the Neuro-ICU with the goal of minimizing the risk of hyper-perfusion injury and/or re-bleed by maintaining close neurologic observation and strict hemodynamics management. The patient’s blood pressure was controlled between 100 and 140 mmHg systolic via IV nicardipine and was monitored and evaluated every hour as part of routine hourly neurologic examinations. In consideration of the hemorrhagic nature of the patient’s presentation, the patient was placed on seizure prophylaxis with Levetiracetam which was administered for 7 post-operative days as there were no clinical seizures noted or detected during this time frame. Other standard post-operative measures taken included a dexamethasone taper to prevent/limit cerebral edema; cefazolin antibiotic prophylaxis as an institutional requirement; DVT prevention measures (early mobilization and use of sequential compression devices with enoxaparin begun on post-op day #1); and maintenance of normoglycemia, normothermia, and stable electrolytes.

The neurological evaluation performed upon admission to the ICU indicated that the patient’s neurological status was the same as it was prior to the surgery; the patient was able to speak fluently and understand what others were saying. Facial symmetry was present, however the patient did have slight flattening of the nasolabial fold on the right side of the face prior to surgery. Pupil size and reaction were both normal, and the patient’s ocular motility was also normal. The patient’s strength in all muscle groups on the left side of his body was rated at MRC 5/5 and in the right upper limb the strength was rated at MRC 4+/5, which was similar to the preoperative findings. No new deficits were identified. During the first post-operative evening, the patient was asymptomatic; he was febrile-free, hypertensive-free, and alert. Speech was fluent and comprehension was precise during multi-step verbal communications. The patient could identify objects, repeat complex sentences, and maintain coherent conversations. There was no indication of dysphasia, dysarthria, or new visual field defects. Initially, when the patient was first mobilized, his gait was steady and he had slightly better fine motor coordination of his right hand than he had preoperatively. The wound dressing was dry and the surgical site was soft, non-tender and without swelling.

For the next 48 h, the patient’s recovery continued uninterrupted. He was eating, ambulating and attending to his own self-care without assistance. The patient’s pain was managed with simple analgesics and there were no episodes of confusion, seizures, or focal neurological decline. The laboratory values returned to normal, and no signs of infection, electrolyte disturbances or venous congestion were identified. The patient reported that he was sleeping well, and described an “ease” of movement and clarity of thought. He stated that the operated hemisphere felt “lighter,” and was more coordinated.

By the third postoperative day, the patient began to read and write spontaneously, and was participating in cognitive exercises with the speech therapy team. The patient demonstrated some subtle improvement in the speed of his speech and naming fluency, likely due to recovery of the cortex after relief of the venous hypertension secondary to the AVM. The cranial suture lines were clean, and the wound edges were beginning to show epithelialization. The patient was fully mobilized on the ward, and was able to converse and walk confidently. A cerebral angiogram immediately post-operatively (Figure 3) confirmed complete removal of the arteriovenous malformation. The antero-posterior and lateral views demonstrated no residual nidus, or early venous filling, and normal cortical perfusion through the middle and posterior cerebral artery distributions. The cortical veins filled symmetrically and drained normally into the superior sagittal and transverse sinuses. The flow pattern was normal, with no evidence of arteriovenous shunt or residual malformation.

The remainder of the patient’s hospital stay was uneventful. The patient’s appetite and sleep returned to normal, and his speech regressed to a normal rhythmic quality. There were no new deficits, and no late complications occurred. The wound healed by primary intention, and the sutures were removed on the tenth postoperative day. The patient was discharged home in good condition, fully oriented, independent in all daily functions, and emotionally stable. At the time of discharge, the patient’s neurological examination was virtually identical to that of before the surgery. Only a mild degree of spasticity of the right hand and a small degree of facial asymmetry persisted. The patient’s modified rankin scale was 1, and his overall recovery was essentially complete in a practical sense. Although the postoperative course was unremarkable from a medical standpoint, it held important implications—the postoperative course represented the resolution not only of the structural abnormality (the malformation), but also the restoration of the relationship between the brain’s structure and function.

Three months following surgery the patient returned for an evaluation. The patient indicated he had a very easy and uneventful recovery at home. Throughout this time frame the patient did not experience any seizures, headaches or fluctuation in neurologic function. In addition, the patients’ energy levels, sleeping and eating habits have all returned to their preoperative states, and he has fully resumed all of his preoperative daily activities with no limitations. The patient is now living independently and has returned to both his professional and social routines as well.

At the time of the three month follow up visit, the patient underwent a thorough neurologic examination which demonstrated a very good and almost complete functional recovery. The patient’s speech was fluent and spontaneous, there were no paraphasic errors, and the patient did not experience any significant delay in finding words. All aspects of comprehension, repetition and reading were intact and naming performance had completely returned to normal. Only minor hypertonicity of the right upper extremity was noted and fine motor coordination had significantly improved with the patient’s handwriting speed being close to its prehemorrhagic baseline. Additionally, facial symmetry had also improved with the patient displaying only a slight residual flattening of the nasolabial area when smiling. Finally, gait was steady and well balanced and the patient did not exhibit any signs of circumduction or instability. All areas of cognitive function including attention, working memory, and executive function were found to be within normal limits. The patient’s modified rankin scale remained 1 and his barthel index was 100. The patient stated that he felt confident in his ability to work and to operate a vehicle safely.

Additionally, a control CT scan was performed at the time of this visit (Figure 4). This study confirmed that the patient’s surgical site had healed well. The post operative cavity located at the left temporo-parieto-occipital junction was clean and clearly defined from the surrounding brain tissue with no evidence of residual vascular structures, hemorrhage or edema. The surrounding brain tissue was noted to be within normal attenuation values and the ventricles and midline structures were intact and unchanged. These findings were consistent with the complete removal of the malformation and restoration of normal cerebral anatomy.

Following this visit, the patient remained clinically stable for months with no development of delayed complications, loss of cognitive function, or recurrence of previous symptoms.

Our goal in this case was to completely remove the arteriovenous malformation with preservation of all the healthy, functioning cortex around it. Our approach was based upon a deep understanding of the anatomy of the brain and the desire to operate within the “natural” planes of the brain, i.e., along the anatomical borders of the brain rather than to display surgical skills. The safety, stability and ability to gradually recover lost function seen in the postoperative course and follow-up provided strong evidence supporting the plan we chose. In our view, the patient presented in this case is not an example of “innovation,” but rather an illustration of what can be accomplished by using an emphasis on structure, sequence and patience. The objective of our surgery was not to demonstrate our technical expertise, but to preserve the physiologic integrity of the injured yet potentially recoverable network. In our view, the clinically and radiographically favorable outcome of this patient supports the premise that carefully executed microsurgery, when performed with an attitude of humility and an appropriate respect for anatomy, can restore balance to areas where normal blood circulation has been interrupted.

The authors wish to present this report as further evidence of the continued value of making decisions about the operative management of arteriovenous malformations based upon anatomy, moving at a deliberate pace during the surgery, and exercising caution throughout the operation.

## 3. Discussion

Neurosurgery continues to be an intellectually challenging and technically complex field—especially when treating cerebral AVMs. Every AVM represents an abnormal vascular relationship, where arteries, veins and brain tissue come together in a way that is both fragile and unique. When considering an AVM, the surgeon does not simply remove an abnormal blood vessel, but restores the blood vessels back to a healthy relationship [13]. In this particular case, we describe a compact, high-flow AVM located at the dominant TPO junction, which is considered one of the most eloquent cortical regions in the brain. The TPO junction is the confluence of three important functional networks: the language network (arcuate fasciculus and posterior superior temporal gyrus), the visual network (Meyer’s loop and lateral occipital cortex), and the multimodal integrative network of the inferior parietal lobule. Each of these functional networks consists of white matter tracts that run very closely to each other, and are separated by only a few millimeters of subcortical tissue. Due to this compact anatomy, the TPO junction is extremely susceptible to any disturbances in the blood supply [14].

When an AVM is located in this area, the vascular anomaly adds to an already densely interconnected network. The continuous high flow of blood through the nidus creates chronic venous hypertension in the surrounding tissue, which impairs capillary perfusion and causes subtle cortical dysfunction long before there is an overt hemorrhage [15]. There have been multiple studies using functional imaging to demonstrate that areas surrounding AVMs are not necrotic, but are instead “stunned”—i.e., in a state of functional dormancy due to circulatory steal. By removing the shunt created by the AVM, the surgeon can reactivate previously suppressed cortical regions—as evidenced by improvements in speech, coordination, and vision. The gradual recovery of the patient’s speech and motor abilities can be attributed to the fact that the resection did not only eliminate the malformation, but also restored perfusion symmetry, thereby allowing the dormant neural circuits to communicate physiologically again [16].

Cerebral AVMs can be treated in three different ways: by means of microsurgical resection, stereotactic radiosurgery (SRS), or endovascular embolization—either independently or sequentially in a multi-modal manner. The selection of the therapeutic modality is usually not algorithmic, but depends on an interaction between angioarchitecture, location, and the functional status of the surrounding cortex [17].

Microsurgical resection of AVMs is the only method that provides immediate and complete nidus eradication, as well as definitive hemodynamic normalization. Modern surgical series report obliteration rates ranging from 96% to 98% for AVMs graded I to III, and favorable neurological outcomes in over 85% of patients [18]. However, surgical risk increases steeply with the grade of the AVM, particularly in eloquent regions, where cortical and subcortical tracts converge in tight anatomical clusters. For example, the TPO junction is an eloquent region, as the boundary between curative resection and irreversible deficit is measured in millimeters [19]. Radiosurgery provides an alternative for deep or surgically inaccessible lesions; however, the latency to cure of radiosurgery is typically between 24 and 36 months, which exposes the patient to ongoing hemorrhagic risk throughout this interval. Additionally, radiosurgery obliteration rates decrease substantially with nidus size and flow rate [20]. Embolization of AVMs via the endovascular route is becoming increasingly sophisticated; nonetheless, it generally serves a complementary purpose. It can reduce shunt flow, eliminate high-risk feeding arteries, and/or serve as a marker for the surgical boundaries of an AVM. However, complete angiographic cure of an AVM solely by means of embolization is rare (typically <40%) [21].

The decision made in this case to perform a direct microsurgical resection of the AVM without previous embolization was based on several factors. The nidus was compact and accessible to the cortex; the arterial and venous structures were easily identifiable, and the gliotic interface between the AVM and the surrounding cortex provided a natural dissection plane [22]. High-flow AVMs pose a significant risk of changing the hemodynamic equilibrium of the AVM with embolization, potentially resulting in intranidal rupture. Therefore, the direct anatomic microsurgical approach was chosen, and is supported by recent evidence indicating that compact high-flow AVMs with defined margins may be best managed by primary resection rather than staged therapy [23].

The TPO junction is an eloquent cortical region of the human brain, consisting of the confluence of three major functional networks: the language network (arcuate fasciculus and posterior superior temporal gyrus), the visual network (Meyer’s loop and lateral occipital cortex), and the multimodal integrative network of the inferior parietal lobule. Each of these networks operate through white matter tracts that lie in close proximity to one another, with only a few millimeters of subcortical tissue separating them. This compact anatomy makes the TPO junction particularly sensitive to any alterations in the blood supply [14]. Therefore, when an AVM is present in this area, the vascular anomaly will add to an already densely interconnected network. Chronic venous hypertension will result from the continuous high flow of blood through the nidus of the AVM, causing impaired capillary perfusion and subtle cortical dysfunction, long before there is an overt hemorrhage. Studies utilizing functional imaging have demonstrated that areas surrounding AVMs do not undergo necrosis; instead, the surrounding cortex becomes “stunned,” and exists in a state of functional dormancy due to circulatory steal [24]. The resection of the AVM shunt can restore previously suppressed cortical regions—a process that can be seen clinically in terms of improved speech, coordination, and vision. The gradual recovery of the patient’s speech and motor capabilities can thus be attributed to the re-establishment of perfusion symmetry, which allowed the dormant neural circuits to once again communicate physiologically [16].

The surgical philosophy for the resection of AVMs differs significantly from those used in the resection of tumors or aneurysms. Rather than simply removing an abnormal circulatory pathway, the surgeon must restore the original physiological circulatory pathway that was disrupted by the AVM. In this type of surgery, the timing and sequence of disconnecting the feeders assume greater importance than the actual dissection itself [18].

The operation described in this case was conducted entirely under the microscope without the aid of neuronavigation, Doppler, or other adjunctive technologies. Each of the feeders was individually identified and dissected to their point of cortical entry, coagulated and then divided under direct visualization. The dissection proceeded in a circumferential manner along the gliotic capsule that represented the historical interface between the AVM and the functioning cortex. This gliotic capsule, which is commonly produced by previous hemorrhages, served as a natural safety margin and facilitated the identification of the normal cortex from the pathological cortex. The intentional preservation of the venous drainage until all arterial inflow was eliminated was a deliberate attempt to prevent postoperative venous infarction and the observation of cortical veins returning to their normal blue coloration indicated successful re-establishment of hemodynamic equilibrium [25]. Due to the compactness of the nidus and minimal disruption of the parenchyma, the entire nidus could be removed as a single unit (en-bloc resection), which is a relatively rare occurrence in high-flow AVMs. Importantly, the operative tempo was slow and deliberate, placing emphasis on the sequence and clarity of the operation over speed. This principle—allowing the anatomy to dictate the rhythm of the operation—is the hallmark of safe AVM surgery and is still relevant today despite the increased availability of technological assistance [26].

The postoperative course of this patient illustrates a fundamental physiological concept: that function in the human brain is not fixed, but can be restored once the hemodynamic environment is normalized. The resolution of venous congestion and the restoration of capillary autoregulation enabled a gradual reactivation of previously suppressed neural pathways. Improvements in speech fluency, motor coordination, and sensory acuity that occurred in the early postoperative period reflect a re-engagement of cortical regions, rather than simple compensation [27]. This phenomenon is known as functional reperfusion recovery and is becoming increasingly recognized as a biological correlate of successful AVM surgery. Recent advances in imaging technology, including perfusion-weighted MRI and resting-state fMRI, have shown that postoperative normalization of local perfusion is associated with improved function, regardless of whether structural changes in the brain have occurred. Therefore, the clinical recovery seen in this case represents not only a surgical success, but also a physiological restoration of network homeostasis [28].

Studies have reported that cerebral AVMs occur at a frequency of 1–1.5 per 100,000 individuals per year, accounting for approximately 3–5% of all intracranial vascular malformations. The majority of AVMs are diagnosed in the third and fourth decades of life and approximately two-thirds of AVMs are diagnosed after a hemorrhagic event. The annual risk of hemorrhage is approximately 2–4% and the cumulative lifetime risk of hemorrhage exceeds 40% by age 40 [29]. Surgically treated AVMs have a mortality rate of 1–3% and a permanent morbidity rate of 5–10%, depending on the grade and location of the AVM. AVMs located at the dominant TPO junction account for less than 5% of resected AVMs, but account for a disproportionately large number of postoperative language and visual deficits. Therefore, even a single successful resection of an AVM in this region will contribute significantly to the understanding of the safe limits of microsurgical resection [30].

Pharmacological therapy cannot directly cause obliteration of the AVM nidus; however, it is crucial for maintaining the stability of the patient during the perioperative period, as well as for providing long-term seizure prophylaxis. Typically, antiepileptic medications are continued for a minimum of six months after surgery because of the heightened excitability of the re-perfused cortex [31]. Corticosteroids may be briefly administered to treat postoperative swelling, but should be discontinued as quickly as possible so as to not interfere with the re-establishment of autoregulatory control. There has been growing interest in the use of anti-angiogenic agents, such as bevacizumab and VEGF-pathway inhibitors for the treatment of inoperable vascular malformations. However, their use for cerebral AVMs is currently speculative and unproven. It is likely that interest in the use of pharmacologic therapies to modify angiogenic signaling will increase in the coming years, and may eventually become a useful adjunct to surgical or radiosurgical treatments of AVMs [32].

Although the case presented was a successful one, there are some limitations to consider. The case represents the treatment of a highly selected lesion in a favorable anatomical configuration, by an experienced surgical team. Therefore, it cannot be generalized to all AVMs, especially larger, more extensive or deeper seated lesions, where the risk of complications may outweigh the benefits of complete resection. Additionally, although the patient’s early and late postoperative imaging demonstrated complete angiographic cure of the AVM, long-term angiographic follow-up is necessary to ensure that there is no recurrence or new development of afferent and efferent channels. In Table 1, we have summarized some previously reported cases and series to give a view of how different authors have approached similar situations and to help put our case into perspective of what has been reported.

More broadly, the limitations of current grading systems, such as the Spetzler–Martin grading system and additional grading systems, are apparent. Although grading systems provide a valuable framework for classification, they fail to take into account the variability of microstructural properties in vascular elastic modulus, venous pressure gradient and regional metabolic reserves, which all affect how surgeons behave and the outcomes of surgery [43]. Integration of advanced functional imaging techniques, including diffusion tensor imaging, tractography, and hemodynamic modeling, into the preoperative evaluation of patients with AVMs represents a promising direction for improving the accuracy of individualized risk assessments. Recent advances in computational neurovascular modeling have enabled simulation of real-time flow redistribution following staged feeder occlusion, providing a glimpse into the potential future of patient specific surgical planning [44]. Additionally, training of artificial intelligence algorithms on angiographic databases may enable these programs to classify AVM angioarchitecture and predict the likelihood of rupture. Nevertheless, even as these technologies evolve, they will be ancillary to, rather than substitutes for, the surgeon’s own anatomical insight—as illustrated in the case described above [45].

Beyond the technical success of the case, the conceptual significance of this case is that it demonstrates that, even in eloquent cortex, curative surgery is possible if the principles of anatomy, timing, and restraint are properly applied. The operation achieved more than a structural correction; it restored a physiological order to a cortical network that was previously compromised by a chaotic hemodynamic environment. Cases like this remind us that the core of neurosurgery is not in the act of intervention, but in the act of restoring function, flow, and harmony in the living brain [46].

## 4. Conclusions

This case has been shared, and so we can take time to appreciate the quiet deliberateness which is still the hallmark of neurosurgeon safety. The complete and accurate removal of a complicated arteriovenous malformation located in the dominant temporo-parieto-occipital junction was accomplished with nothing new or innovative, but through careful observation of anatomy, deliberate pace and restraint. Each and every action taken during surgery was performed in accordance with the normal anatomical order of the brain itself; therefore, the procedure took place as if the surgeon were having a conversation with the living tissue of the brain rather than as if he were engaged in some sort of competition against the living tissue of the brain.

The patient’s gradual recovery from surgery serves as a reminder that while surgery is meant to remove the pathological process, its ultimate goal is to restore the physiological rhythms within the body’s networks, which have become disordered. Therefore, the “cure” in this case did not occur at the moment of pathological removal, but occurred in the form of the gradual return of the patient’s use of language, movement and overall self-confidence that occurred after surgery, demonstrating the brain’s potential to find its own way back into balance when treated with respect by a neurosurgeon.

At a time when the practice of microsurgery is continuing to evolve with the advent of new technologies such as advanced imaging and computational models, this case is a gentle reminder that there are certain aspects of the practice of surgery which cannot be replaced by these technologies, including experience and the intuitive knowledge gained by carefully listening to the anatomy of the brain. While the technology used to aid in the practice of microsurgery will continue to evolve, one aspect of the practice of microsurgery will remain unchanged—the quiet discipline of understanding what should be left intact.

Therefore, this report will not serve as a conclusion but as a pause—a time of gratitude for the brain’s ability to heal, and for the privilege of being able to witness that healing through the microscope.

## Figures and Tables

**Figure 1 diagnostics-15-03249-f001:**
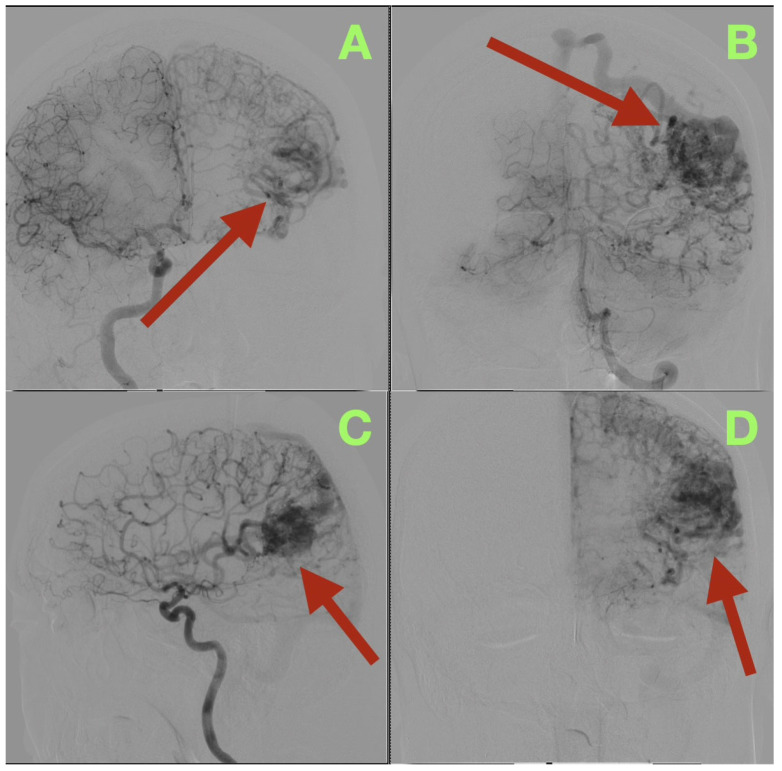
Preoperative digital subtraction angiography (DSA). (**A**): Lateral projection during selective left internal carotid injection demonstrates a compact, high-flow nidus (arrow) centered at the temporo-parieto-occipital junction. The arterial convergence from distal MCA and PCA branches is clearly visualized, with early venous filling consistent with a high shunt volume. (**B**): Anteroposterior projection highlights the dense arteriovenous network occupying the posterior perisylvian region (arrow). Multiple feeders from M4, P4, and A4 segments converge toward the nidus, producing an intricate lace-like configuration typical of compact high-flow AVMs. (**C**): Late arterial phase delineates the dual venous drainage (arrow), with a superior cortical outflow ascending to the superior sagittal sinus and a secondary temporobasal collector directed toward the transverse-sigmoid complex. (**D**): Venous phase confirms synchronous opacification of both drainage routes (arrow), demonstrating the bidirectional flow pattern and the absence of venous outflow restriction. The preserved parenchymal perfusion surrounding the nidus suggests a stable hemodynamic equilibrium between the AVM and eloquent cortical territories.

**Figure 2 diagnostics-15-03249-f002:**
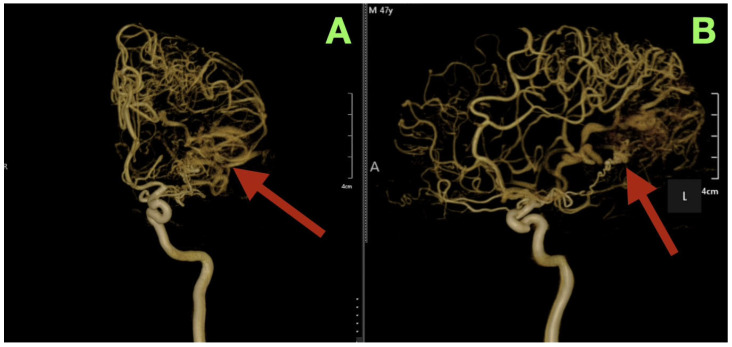
Three-dimensional rotational angiography reconstructions. (**A**): Left lateral oblique view showing the compact nidus (arrow) embedded beneath the posterior superior temporal and angular gyri. The close spatial relationship to the arcuate fasciculus and Meyer’s loop explains the patient’s expressive aphasia and right inferior quadrantanopia. (**B**): Posterior oblique projection illustrating the deep extension of the nidus toward the occipital surface (arrow) and the tangential entry of short pial feeders from M4 and P4 branches. The surrounding venous system demonstrates smooth, ectatic channels draining into both the superior sagittal and transverse-sigmoid sinuses, confirming a mixed superficial-deep outflow pattern.

**Figure 3 diagnostics-15-03249-f003:**
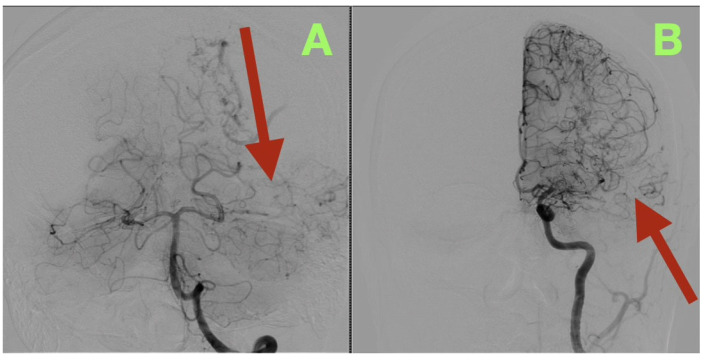
Immediate postoperative digital subtraction angiography. (**A**): Frontal projection showing complete exclusion of the previously identified left temporo-parieto-occipital arteriovenous malformation (arrow). No early venous opacification or residual nidus is seen. (**B**): Lateral projection demonstrating normal cortical circulation and venous drainage through the superior sagittal and transverse sinuses (arrow). The vascular architecture is preserved and physiological, confirming complete anatomic and hemodynamic cure.

**Figure 4 diagnostics-15-03249-f004:**
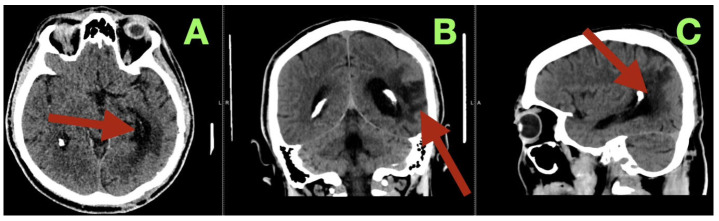
Three-month postoperative CT control scan. (**A**): Axial non-contrast CT image shows a well-defined postoperative cavity at the left temporo-parieto-occipital junction (arrow) with no evidence of residual vascular structures, hemorrhage, or ischemia. The surrounding cortex demonstrates normal attenuation, and there is no perilesional edema or mass effect. (**B**): Coronal reconstruction reveals a narrow linear gliotic tract corresponding to the prior surgical corridor (arrow), with preserved cortical and subcortical architecture. Ventricular size and midline structures are normal. The scan confirms durable anatomical restitution of the operated hemisphere, absence of recurrence, and satisfactory long-term healing. (**C**): Sagittal reconstruction demonstrates the postoperative resection cavity as a sharply demarcated hypodense region (arrow) along the prior surgical trajectory, with smooth margins and mature gliotic lining.

**Table 1 diagnostics-15-03249-t001:** This table summarizes pivotal clinical and translational studies that define the modern surgical philosophy for cerebral AVMs located at the TPO junction. It intends to integrate historical grading principles, contemporary microsurgical outcomes, physiological recovery mechanisms, and computational modeling insights that together inform a hemodynamically guided, anatomy-respecting approach. Collectively, these studies illustrate how precise timing, flow restoration, and functional restraint remain central to achieving curative yet safe AVM resection in eloquent cortical regions.

Author(s) and Year	Study Type	Population	Key Findings	Relevance to This Case
Spetzler & Martin (1986) [33]	Prospective classification study	100 cerebral AVM cases	Introduced the Spetzler–Martin grading system correlating AVM size, venous drainage, and eloquence with surgical risk.	Provides the grading framework supporting risk stratification and operative decision-making for TPO AVMs.
Lawton et al. (2010) [34]	Retrospective surgical series	250 patients with AVMs (grades I–III)	Reported 96–98% obliteration and 85% favorable outcomes in eloquent regions with anatomy-guided microsurgery.	Validates microsurgical cure in compact, high-flow AVMs within eloquent cortex under precise anatomical control.
Schramm et al. (2017) [35]	Multicenter cohort	288 eloquent AVMs	Demonstrated that meticulous disconnection of arterial feeders and delayed venous division minimize infarction risk.	Supports the stepwise feeder-first, vein-last dissection technique applied in this case.
Abla et al. (2014) [36]	Prospective registry	343 AVM resections	Early surgical intervention improved functional outcomes and reduced cumulative hemorrhagic risk.	Reinforces early resection in accessible, high-flow lesions to prevent recurrent hemodynamic injury.
Bigder et al. (2021) [37]	Outcome analysis	176 eloquent cortical AVMs	“Anatomy-paced” microsurgery achieved >90% functional preservation, emphasizing gliotic plane dissection.	Corroborates the slow, deliberate dissection tempo used to maintain eloquence at the TPO junction.
Hernesniemi et al. (2008) [38]	Prospective single-center	238 AVMs (eloquent & deep)	92% angiographic cure, 5% permanent morbidity; compact, superficial AVMs fared best.	Confirms favorable outcomes in compact AVMs like this TPO lesion with clear cortical interface.
Potts et al. (2016) [39]	Functional neuroimaging study	42 patients with AVMs	Demonstrated “stunned cortex” phenomenon—perilesional areas recover function after flow restoration.	Explains postoperative recovery of speech and coordination following resection and reperfusion.
Jo et al. (2011) [40]	Multimodal imaging correlation	22 AVM cases	Perfusion-fMRI revealed normalization of local flow correlates with improved neurological scores postoperatively.	Supports the physiological rationale for functional reperfusion recovery seen in this patient.
Zhu et al. (2025) [41]	Computational hemodynamic modeling	60 AVMs analyzed	Simulations showed that resection of dominant feeders restores perfusion balance and lowers rupture potential.	Reinforces hemodynamic reasoning behind direct surgical correction rather than staged embolization.
Sattari et al. (2023) [42]	Systematic review	412 cases, eloquent-region AVMs	Microsurgery remains the gold standard for compact cortical AVMs, with radiosurgery reserved for deep/inoperable lesions.	Aligns with the surgical choice for a compact, accessible nidus at the TPO junction.

## Data Availability

The data presented in this study are available on request from the corresponding author.

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
