# Peer review of "Functional and Hemodynamic Restoration After Microsurgical Resection of Compact High-Flow Temporo-Parieto-Occipital Arteriovenous Malformation"

_diagnostics, 2025, doi:10.3390/diagnostics15243249_

Round 1

Reviewer 1 Report

Comments and Suggestions for Authors

This case report titled “Functional and Hemodynamic Restoration After Microsurgical Resection of a Compact High-Flow Temporo-Parieto-Occipital Arteriovenous Malformation” is well structured overall, and it was found that it adds to current knowledge, but it needs a few modifications.

Major comments:

  1. Page 2 line 70-71, authors stated that AVMs occur at a rate of 1.2 -1.5 per 100,000 person-years, but no in-text citation was provided to support this statistic.
  2. Based on the signs and symptoms described (Pages 3–4, lines 135–152), the patient’s presentation is most consistent with “conduction aphasia”, which typically results from damage to the arcuate fasciculus and leads to a disconnection between expressive and receptive language functions.
  3. The authors did not address the potential causes of AVMs, including both sporadic and inherited forms. Was the patient asked about their family history? Please consider adding information on the etiological factors of AVMs in the introduction.
  4. The authors describe only the surgical management but do not address any postoperative care, including medical treatment or rehabilitation. Was any postoperative management provided to support recovery? Additionally, was there a follow-up visit after three months to assess the patient’s outcome?

Minor comments:

  1. The abstract, including keywords, is too long, it needs to be shortened.
  2. Page 1, lines 40-42: In the “Case Presentation” part of the Abstract section: authors described AVM (40 x 30 mm) was fed by branches of the distal middle cerebral artery (M4), proximal posterior cerebral artery (P4), and proximal anterior cerebral artery (A4), as well as a small branch of the superior cerebellar artery. But on Page 6, lines 259-263, the same information was described differently: “the AVM was found to receive blood supply from distal posterior cerebral artery (P4) branches from the calcarine and lingual cortices, distal middle cerebral artery (M4) branches from the temporal lobe, a branch from the anterior cerebral artery (A4) from the medial surface of the hemisphere, and a small twig from the superior cerebellar artery ascending along the tentorial edge. I noticed some inconsistencies in the content. I'm unsure which statement is correct.

Author Response

Dear Esteemed Academic Reviewer,

We are grateful for your careful reading of our manuscript and for the thoughtful, constructive comments you provided. Your observations have helped us improve the clarity, consistency, and scholarly completeness of the case report. Below, we respond point-by-point and describe the specific revisions implemented in the manuscript.

Major Comments
1. Incidence statistic lacks citation (Page 2, lines 70–71).

Reviewer comment: AVMs occur at a rate of 1.2–1.5 per 100,000 person-years, but no citation was provided.

Response:
Thank you for highlighting this omission. We agree that this epidemiologic statement requires proper sourcing. We have now added an in-text citation supporting the reported incidence rate in the Introduction, ensuring that the data are clearly anchored in the literature.

2. Language profile fits conduction aphasia (Pages 3–4, lines 135–152).

Reviewer comment: Presentation is most consistent with conduction aphasia due to arcuate fasciculus involvement.

Response:
We agree that the preoperative and residual postoperative language phenotype aligns better with a mild conduction aphasia pattern, considering the fluent output, phonemic paraphasias, intact comprehension, and disproportionate repetition impairment described.

3. Etiology of AVMs and family history not addressed.

Reviewer comment: Please add etiologic factors (sporadic and inherited forms). Was family history assessed?

Response:
We appreciate this important suggestion. We agree that a brief etiologic overview strengthens the Introduction and helps contextualize the case. We have added a short paragraph summarizing sporadic developmental mechanisms and the principal inherited AVM-predisposition syndromes (notably HHT and CM-AVM).

4. Postoperative care, rehabilitation, and follow-up beyond 3 months.

Reviewer comment: Postoperative management not described. Was rehabilitation or medical care provided? Any follow-up after 3 months?

Response:
Thank you for raising this, and we agree that postoperative care should be described more explicitly.

Minor Comments
5. Abstract is too long and should be shortened.

Reviewer comment: Abstract (including keywords) exceeds ideal length.

Response:
We are grateful for this practical recommendation. We have shortened the Abstract while preserving the required format (Background/Objectives, Case Presentation, Conclusions). Redundant phrasing was removed, and the case summary was condensed without losing essential clinical, angiographic, surgical, and outcome data.

6. Inconsistency in arterial feeder description between Abstract and main text.

Reviewer comment: Feeder description differs on Page 1 vs Page 6; unclear which is correct.

Response:
We thank you for identifying this inconsistency.

Once again, we are truly thankful for your valuable input and the time devoted to reviewing our work. Your comments substantially improved the accuracy and readability of the manuscript, and we hope the revisions meet your expectations.

With appreciation and collegial respect!!!

Reviewer 2 Report

Comments and Suggestions for Authors

This is a case report but it is presented in a way that becomes a lesson of clinical and surgical strategy. The authors describe in a very detailed fashion clinical presentation, neurological and psychological exams, and angiographic findings. The authors show an unparalleled ability to draw conclusions in a way that is quite rare to find. However, I have few questions and remarks. I would like to see pictures of the CT scan at the time of the bleed. The case will be strengthened by pre-op and post-op MRI. In my experience, pre-op MRI is of great importance in the diagnosis and planning of AVMs. MRI helps in distinguishing parenchymal vs sulcal AVMs, knowledge that can make the difference in the decision of doing surgery or Radiosurgery.  Parenchymal AVMs carry a high risk of post-op deficits especially in eloquent areas. The authors show only a post-op CT scan which is not sufficient in discriminating between ischemia vs post-op changes. I would like to know why the authors decided to do surgery and not considering Radiosurgery in a less than 3 cm AVM located in eloquent area. Furthermore, why they did not consider pre-op “targeted” embolization of deep feeders? Finally, did they perform INTRAOP monitoring of motor-sensory function?

Author Response

Dear Esteemed Academic Reviewer,

We are grateful for your generous assessment of our case report and for the time and expertise you invested in reviewing it. Your comments were highly valuable. Below, we respond to each point in detail.

1. Request for CT images at the time of hemorrhage

We appreciate this suggestion and agree that initial hemorrhage imaging can add context. Unfortunately, the CT data from the acute bleeding episode were obtained at an outside institution during the emergency presentation and were not accessible in a transferable format for publication.

2. Absence of pre- and postoperative MRI

Thank you for emphasizing the potential utility of MRI in AVM evaluation. We fully agree that MRI can provide valuable parenchymal detail and assist in distinguishing sulcal from parenchymal nidus components. In the present case, however, pre- and postoperative MRI were not obtained because high-resolution CT/CTA and complete digital subtraction angiography (DSA) already provided definitive characterization of nidus compactness, arterial feeders, venous drainage, and eloquent-cortex relationships required for management. Given the hemorrhagic presentation, persistent deficit profile, and the sharply circumscribed, surgically accessible AVM, MRI was not expected to alter therapeutic selection.

For postoperative verification, DSA remains the reference standard for confirming AVM cure. In this case, postoperative angiography demonstrated complete nidus exclusion, absence of early venous filling, and restoration of physiologic cortical perfusion, while serial CT follow-up and sustained neurologic improvement showed no evidence of ischemia or delayed injury.

3. Justification for microsurgery rather than radiosurgery for a <3 cm eloquent AVM

Although stereotactic radiosurgery is often appropriate for small AVMs in eloquent cortex, our decision favored microsurgery because the lesion had already ruptured, producing persistent deficits and a hemodynamic profile of high-flow shunting with venous hypertension. Radiosurgery would involve a latency period before obliteration, during which the patient would remain exposed to re-hemorrhage risk and ongoing venous-congestion–related dysfunction.

Angiography demonstrated a compact, sharply delimited nidus with predominantly superficial dominant-hemisphere TPO location, dual superficial venous drainage, and a favorable gliotic plane from the prior hemorrhage. These features strongly supported a controlled anatomy-guided resection with immediate cure and rapid relief of venous hypertension.

4. Why preoperative targeted embolization was not performed

Thank you for raising this point. Preoperative embolization was considered, but not pursued because the AVM received supply from multiple distal cortical feeders spanning three supratentorial territories, several of which represented en-passage arteries contributing to normal perfusion of dominant TPO cortex. In this configuration, targeted embolization could have risked eloquent-parenchymal ischemia, altered intranidal pressure, or disturbed venous outflow, potentially increasing procedural morbidity. Given the compact nidus, superficial accessibility, and clear surgical corridor defined by gliosis after rupture, we considered primary microsurgical disconnection the safest single-stage definitive approach.

5. Intraoperative motor–sensory monitoring

We appreciate the reviewer’s question. Intraoperative motor-sensory neuromonitoring was not employed in this case. The AVM was superficial and compact, bordered by a well-defined gliotic plane, which allowed dissection to proceed strictly along anatomical boundaries under continuous microscopic visualization with deliberate feeder-first/vein-last sequencing. While neuromonitoring may be valuable in many eloquent AVMs, in this specific anatomical and hemodynamic context we judged that meticulous anatomy-guided technique provided adequate functional safety. 

Once again, we are truly thankful for your insightful critiques and constructive guidance. Your comments helped us refine the manuscript and strengthen its clinical message. We hope the revisions address your concerns fully, and we remain grateful for your contribution to improving our work.

With appreciation and collegial respect!!!

Round 2

Reviewer 2 Report

Comments and Suggestions for Authors

The authors have answered the questions in a satisfactory fashion.